# PI3K–AKT Signaling Activation and Icariin: The Potential Effects on the Perimenopausal Depression-Like Rat Model

**DOI:** 10.3390/molecules24203700

**Published:** 2019-10-15

**Authors:** Li-Hua Cao, Jing-Yi Qiao, Hui-Yuan Huang, Xiao-Yan Fang, Rui Zhang, Ming-San Miao, Xiu-Min Li

**Affiliations:** 1International TCM Immunopharmacology Research Center, Henan University of Chinese Medicine, Zhengzhou 450046, China; caolihua727@126.com (L.-H.C.); qiaojingyi618@126.com (J.-Y.Q.); 2Department of Pharmacology, Henan University of Chinese Medicine, Zhengzhou 450046, China; yaolihuanghuiyuan@126.com (H.-Y.H.); fxylele@yeah.net (X.-Y.F.); zhangruiyaoli@126.com (R.Z.); 3Microbiology and Immunology Department, New York Medical College, Valhalla, NY 10595, USA

**Keywords:** icariin, perimenopausal depression, PI3K–AKT signaling, rats, forced swimming test, open field test

## Abstract

Icariin is a prenylated flavonol glycoside isolated from Epimedium herb, and has been shown to be its main bioactive component. Recently, the antidepressant-like mechanism of icariin has been increasingly evaluated and demonstrated. However, there are few studies that have focused on the involvement of the phosphatidylinositol 3-kinase (PI3K)/serine-threonine protein kinase (AKT) signaling in mediating the perimenopausal depression effects of icariin. Perimenopausal depression is a chronic recurrent disease that leads to an increased risk of suicide, and poses a significant risk to public health. The aim of the present study was to explore the effect of icariin on the expression of the PI3K–AKT pathway related to proteins in a rat model of perimenopausal depression. Eighty percent of the left ovary and the entire right ovary were removed from the model rats. A perimenopausal depression model was created through 18 days of chronic unpredictable stimulation, followed by the gavage administration of target drugs for 30 consecutive days. We found that icariin administered at various doses significantly improved the apparent symptoms in the model rats, increased the organ indices of the uterus, spleen, and thymus, and improved the pathological changes in the ovaries. Moreover, icariin administration elevated the serum levels of female hormone estradiol (E_2_), testosterone (T), and interleukin (IL)-2, decreased those of follicle stimulating hormone (FSH) and luteotropic hormone (LH), promoted the expression levels of estrogen receptor (ER) and ERα in the hypothalamus, and increased those of serotonin (5-HT), dopamine (DA), and noradrenaline (NA) in the brain homogenate. Furthermore, icariin elevated the expression levels of AKT, phosphorylation-akt (p-AKT), PI3K (110 kDa), PI3K (85 kDa), and B-cell lymphoma 2 (Bcl-2) in the ovaries, and inhibited those of Bax. These results show that icariin administration rebalanced the disordered sex hormones in perimenopausal depression rats, regulated the secretion of neurotransmitters in the brain, boosted immune function, and improved the perimenopausal syndrome. The mechanism of action may be related to the regulation of the expression of PI3K–AKT pathway-related proteins.

## 1. Introduction

Perimenopausal depression is a mental disorder that first occurs in women during the perimenopausal period and is mainly characterized by sleep disorders, sexual concerns, tiredness, weight changes, anxiety, nervousness, loss of interest, and deterioration in memory and concentration, especially of the ovarian function [1,2]. Women with severe symptoms may have a tendency to commit suicide. Psychosocial factors, including major life events, children who leave home, and negative attitudes against aging have been reported as risk factors for perimenopausal depression [3,4,5]. Between ages 42 and 55, nearly all women experience the menopause transition; the perimenopause is considered as a critical period for the development of depression [6]. The implications of the perimenopausal symptoms for depression merit attention, because depression poses a substantial burden on the well-being of women and their families, particularly during midlife [7]. Perimenopausal depression is caused by serum levels of estradiol fluctuating during the menopausal transition and psychosocial factors [8]. A study has shown that perimenopausal estradiol fluctuation increases sensitivity to psychosocial stress and vulnerability to depressed mood [9,10]. Women who have more symptoms of depression in their early 40 s may be at heightened risk for problems with the menopausal transition. Conversely, efforts to address more severe symptoms of menopause may help to reduce the onset of depressive symptoms among middle-aged women [6]. Estrogen exerts antidepressant effects in some perimenopausal women [11,12], but its use for perimenopausal depression is not Food and Drug Administration (FDA) approved [13]. Moreover, the wide variety of side effects limits its clinical application [14]. There is no doubt that treating depression during perimenopause requires an effective and safe antidepressant.

Therefore, there are many depressive patients, some with the belief that “natural is better”, and treating perimenopausal depression with Chinese herbal medicine has become a research hotspot [15,16]. Phytoestrogen, as a large group of heterocyclic phenolic compounds with chemical structure similar to that of estradiol, has received intense attention. Icariin (2-(4′-methoxylphenyl)-3-rhamnosido-5-hydroxyl-7-glucosido-8-(3′-methyl-2-butylenyl)-4-chromanone) is a prenylated flavonol glycoside monomer extracted from Herba Epimedii (Yin Yang Huo) that is classified as a kidney-yang tonifying medicinal herb in traditional Chinese medicine (TCM). It is also available as a dietary supplement in the United States [17]. Icariin exerts various pharmacological effects, including regulating immunity and promoting the proliferation and development of osteoblasts [18], which exhibit excellent medicinal and health benefits. Icariin has been demonstrated to protect against diseases associated with estrogen deficiency [19]. Previous studies have demonstrated that icariin exerts an antidepressant effect in an unpredictable chronic mild stress model of depression in rats [20]; however, the effect and underlying mechanism of icariin in the treatment of perimenopausal depression has not yet been elucidated.

The PI3K–AKT signaling pathway is involved in the proliferation, differentiation, and migration of ovarian granulosa cells. On one hand, this pathway regulates cell proliferation and division, and on the other hand, it controls cell apoptosis. Currently, research regarding the PI3K–AKT pathway is primarily focused on cardio-cerebral-vascular disease, tumorigenesis [21], ovarian dysplasia, and breast cancer [22], and has not yet been applied to perimenopausal syndrome studies. The fundamental cause of perimenopausal depression resides in the decline of ovarian function, while ovarian cell proliferation and apoptosis are important factors affecting ovarian function. Nevertheless, studies have reported that icariin may promote the osteogenic differentiation of bone marrow stromal cells and osteoblasts associated with estrogen deficiency by activating the PI3K–AKT pathway [23]. Furthermore, another study showed that icariin exerted stronger anti-apoptotic effects than 17β-estradiol, and inhibited the cleavage of downstream caspase-3 in MC3T3-E1 cells induced by a potent PI3K inhibitor [24]. However, antidepressant treatment of icariin on PI3K–AKT signaling is still seldom studied in a chronic unpredictable mild stress (CUMS) model of animals. Therefore, the present study was designed to study whether the flavonoid compound isolated from Herba Epimedii, namely icariin, exerts an antidepressant-like effect in a perimenopausal depression model by activation of the PI3K–AKT pathway, and to provide objective experimental evidence for the potential clinical application of icariin in the future.

## 2. Results

### 2.1. Effects of Icariin on Behavioral Changes in the Perimenopausal Depression Rats

To examine the effects of icariin, we used the open field test as a well-established paradigm to measure locomotion and depression-like behavior in animals. As shown in Figure 1A,B, Compared with the sham group (SG), the movement distance and stand-up times in the model group (MG) were significantly decreased (*p* < 0.01). It showed that the activity ability, cognitive ability, and the curiosity about the fresh environment (the ‘willingness’ to explore the environment) of the model group rats were decreased. Compared with the MG, the movement distance and stand-up times in the estradiol valerate group (EVG) and high-dose icariin group (IG-HD) were significantly induced (*p* < 0.01). In the medium-dose icariin group (IG-MD), the movement distance and stand-up times were clearly higher (*p* < 0.05). In the low-dose icariin group (IG-LD), the stand-up times were significantly elevated (*p* < 0.05), and the movement distance demonstrated a rising trend (*p* > 0.05).

The forced swimming test is a stress-induced escape reduction paradigm that is largely used to measure depression-like behavior. Immobility was calculated as the length of time meant to perform the minimum movement necessary to stay afloat. Based on the data shown in Figure 1C, compared with the SG, the immobility time in the MG was significantly increased (*p* < 0.01). It showed that the degree of despair to the adverse environment of the model group rats was increased. Compared with the MG, the immobility time in the EVG, IG-HD, and IG-MD were decreased significantly (*p* < 0.01). In the IG-LD, the stand-up times were significantly elevated (*p* < 0.05), while the immobility time was decreased (*p* < 0.05), and the movement distance demonstrated a rising trend (*p* > 0.05).

### 2.2. Effects of Icariin on Serum Biochemical Indicators in the Perimenopausal Depression Rats

Figure 2 shows the effects of icariin on the serum levels of estradiol (E_2_), follicle stimulating hormone (FSH), luteotropic hormone (LH), testosterone (T), and interleukin (IL)-2 in perimenopausal depression rats. Compared with the SG, the serum levels of E_2_, IL-2, and T in the MG were significantly decreased (*p* < 0.01), while those of LH and FSH were significantly induced (*p* < 0.01), which is consistent with hormonal changes in perimenopausal depression. Compared with the MG, the serum levels of E_2_, IL-2, and T in the EVG were significantly increased (*p* < 0.01), while those of LH and FSH were decreased significantly (*p* < 0.01). In the IG-HD, the E_2_, T, and IL-2 levels were clearly higher (*p* < 0.05), while the FSH levels were significantly lower (*p* < 0.01) and the LH levels were clearly reduced (*p* < 0.05). In the IG-MD, the serum T levels were significantly higher (*p* < 0.01), and the E_2_ and IL-2 levels were clearly higher (*p* < 0.05), while the LH levels were significantly lower (*p* < 0.01) and the FSH levels were clearly reduced (*p* < 0.05). In the IG-LD, the serum IL-2 levels were significantly elevated (*p* < 0.05), the LH levels were clearly decreased (*p* < 0.05), and the FSH levels demonstrated a declining trend (*p* > 0.05), while the E_2_ and T levels demonstrates a rising trend (*p* > 0.05).

### 2.3. Effects of Icariin on Serum Biochemical Indicators in the Perimenopausal Depression Rats

The findings presented in Figure 3 are quite revealing in several ways. Compared with the SG, the brain homogenates levels of serotonin (5-HT), dopamine (DA), and noradrenaline (NE) in the MG were significantly suppressed (*p* < 0.01). It showed that the neurotransmitter secretion level of the perimenopausal depression model rats was in disorder. Compared with the MG, the brain homogenates levels of 5-HT, DA, and NE in the EVG was significantly augmented (*p* < 0.01). In the IG-HD, the brain homogenates levels of DA and NE were significantly higher (*p* < 0.01), and the 5-HT levels were clearly higher (*p* < 0.05). In the IG-MD, the brain homogenates levels of 5-HT and NE were significantly higher (*p* < 0.01), and the DA levels were clearly higher (*p* < 0.05). In the IG-LD, the levels of 5-HT and NE were clearly enhanced (*p* < 0.05), and the DA levels demonstrated a rising trend (*p* > 0.05).

### 2.4. Effects of Icariin on Visceral Tissue in the Perimenopausal Depression Rats

As shown in Figure 4, compared with the SG, the thymus, spleen, and uterine indexes in the MG were significantly decreased (*p* < 0.01). It is basically consistent with the pathological changes of organs in perimenopausal depression. Compared with the MG, the thymus, spleen, and uterine indexes in the EVG and IG-HD were significantly increased (*p* < 0.01). The thymus and spleen indexes in the IG-MD and the IG-LD were significantly increased (*p* < 0.01), and the uterine index was clearly augmented (*p* < 0.05).

### 2.5. Effects of Icariin on the Pathological Changes of Ovary in the Perimenopausal Depression Rats

As shown in Figure 5, sections from SG rats revealed no histopathological alterations and normal histological structures of ovaries (Figure 5B). On the other hand, MG rats showed no folliculi in the ovaries, a lot of corpus luteum, ovarian atrophy, and no blood vessels (Figure 5C). After treatment with icariin, the damage to the ovaries was alleviated. More interestingly, its efficacy is comparable to that of estradiol valerate. EVG (Figure 5D) and IG-HD (Figure 5E) rats showed that the growing ovarian follicle, mature ovarian follicle, and corpus luteum were visible in the ovarian tissue. Moreover, IG-MD (Figure 5F) and IG-LD (Figure 5G) rats showed that the mature ovarian follicle, corpus luteum, and the number of layers of granulosa cells was less. These results are also presented in Figure 6A; by Ridit test, compared with the SG, the ovaries of the model group showed significant pathological changes (*p* < 0.01). Estradiol valerate and each dose of icariin can ameliorate the ovary pathological changes in the perimenopausal depression of rats (*p* < 0.01).

### 2.6. Effects of Icariin on the Expression of Hypothalamus ER and ERα in the Perimenopausal Depression Rats

Based on the data shown in Figure 6, in the hypothalamus, using a light microscope to observe the results, positive staining shows different degrees of yellow or pale brown (Figure 7C–N). The calculation of the average optical based on the results shows that compared with the SG, the expression of estrogen receptor (ER) and ERα in the ovaries of MG rats was decreased significantly (*p* < 0.01). Compared with the MG, the expression of ER and ERα in the hypothalamus of the EVG and IG-HD were significantly increased (*p* < 0.01). The expression of ER in the IG-MD rats was significantly increased (*p* < 0.01), and the expression of ERα was clearly increased (*p* < 0.05). In the IG-LD, the ER and ERα expression were clearly elevated (*p* < 0.05).

### 2.7. Effects of Icariin on the Expression of Ovary Bax and Bcl-2 in the Perimenopausal Depression Rats

Figure 7 shows the effects of icariin on the ovaries expression of bax and B-cell lymphoma 2 (Bcl-2) in perimenopausal depression rats. In the ovaries using a light microscope to observe the results, positive staining shows different degrees of yellow or pale brown. Calculation of the average optical based on the results shows that compared with the SG, the expression of Bcl-2 in the ovaries of model rats was inhibited significantly (*p* < 0.01), while the expression of Bax was increased significantly (*p* < 0.01). Compared with the MG, the expression of Bcl-2 in the ovaries of the EVG and IG-HD was significantly improved (*p* < 0.01), and the expression of Bax was suppressed significantly (*p* < 0.01). In the IG-MD, the expression of Bcl-2 was increased (*p* < 0.01), and the expression of Bax was significantly decreased (*p* < 0.01). In the IG-LD, the bax expression was clearly decreased (*p* < 0.05).

### 2.8. Effects of Icariin on the Expression of Ovaries p-AKT/AKT, PI3K (110 kd) and PI3K (85 kd) in the Perimenopausal Depression Rats

As shown in Figure 8, compared with the SG, the P-AKT/AKT and expression of PI3K (110 kd) and PI3K (85 kd) in the ovaries of model rats were attenuated significantly (*p* < 0.01). Compared with the MG, the P-AKT/AKT and expression of PI3K (110 kd) and PI3K (85 kd) in the ovaries of the EVG and IG-HD were significantly promoted (*p* < 0.01). In the IG-MD, the PI3K (110 kd) and PI3K (85 kd) expression were significantly higher (*p* < 0.01). In the IG-LD, the P-AKT/AKT and expression of PI3K (110 kd) were significantly elevated (*p* < 0.01), and the PI3K (85 kd) expression was clearly increased (*p* < 0.05).

## 3. Discussion

Herbal medication, one of the complementary and alternative medicine options, is currently popular in China. Herba Epimedii has been used to treat perimenopausal depression in the clinic for a long time. Icariin, a flavonoid and a major constituent of Herba Epimedii, has been previously demonstrated to possess potential antidepressant-like effects [25]. It attracted our attention for further development. A large number of studies [26,27,28,29,30,31] demonstrated the efficacy of estrogen in perimenopausal and perimenopausal depression. In women with past perimenopausal depression who were previously responsive to hormone therapy, the recurrence of depressive symptoms during hormone withdrawal suggests that normal changes in ovarian estradiol secretion can trigger an abnormal behavioral state [32]. A longer duration of estrogen exposure from menarche to the onset of menopausal transition was found to be significantly associated with a lower risk of depression [33]. Studies indicated that hormone replacement therapy (HRT) increased postmenopausal women’s risk of developing breast cancer, stroke, thrombosis, and cardiovascular disease [34,35], but there is no doubt that estrogen continues to be the most effective treatment to relieve perimenopausal depression [36,37]. Thus, we used estradiol valerate tablets as positive control to determine the efficacy of icariin in the treatment of perimenopausal depression.

The perimenopausal depression model is a complex model that combines a perimenopausal syndrome model with a depression model. Generally, animal depression models can be divided into three types: the chronic stress model, the solitary model, and the olfactory tubercle excision model. Among these, the chronic stress model is the most commonly used. Removal of the ovaries in rats or mice results in a substantial reduction in the supply of ovarian-derived estrogens, simulating the hormonal changes in perimenopausal humans. The chronic unpredicted stimuli simulate various difficulties encountered in life. The present animal model caused changes in neurotransmitters and emotions similar to perimenopausal depression and simulated the key features of endocrine disorders in perimenopausal syndrome; therefore, this was an ideal model for perimenopausal depression [38]. Some studies [24,39,40,41] were carried out on the antidepressant-like effects of drug on a model of depression produced by CUMS for 28 days in long-term ovariectomized adult rats. Perimenopausal women have complete ovaries, and their hormone levels showed a gradual decline. Meanwhile, ovariectomized animal model hormones are suddenly declining, making the model different from human diseases. Reserving part of the ovaries in this study can better simulate the slow decrease of estrogen in women [38].

In the present study, we established a rat model of perimenopausal depression induced by CUMS and a part of the ovary was removed from the rat in order to examine the effects of icariin treatment. To verify whether icariin possesses the antidepressant-like effects in perimenopausal animals, we used an open field test and forced swimming test. The present study found that 30 days of administration of icariin significantly increased the stand-up times and movement distance in the open field test, and decreased the immobility time in the forced swimming test in perimenopausal rats exposed to CUMS, thus confirming the therapeutic action of icariin in an experimental setting, which clearly suggested the antidepressant-like effects of icariin in perimenopausal depressive-like rats.

The pathogenesis of perimenopausal depression includes unbalanced neuroendocrinology, apoptosis, changes in immune function, effects of vasomotor factors and free radicals, decline in immunity, and changes in the monoamine neurotransmitter levels in the brain. Estrogens (mainly estradiol, E_2_) and T are steroid hormones secreted by the ovaries that promote the growth and maturation of the female reproductive viscera. FSH stimulates the maturation and growth of follicles, and synergistically, LH stimulates the ovaries to secrete E_2_. Upon maturation of the follicles, FSH and LH reach their peak values. During the perimenopausal period, ovarian function declines, estrogen levels decrease, and the pituitary gland promotes the secretion of FSH and LH due to the lack of feedback effect of E_2_. The changes in E_2_, FSH, and LH are also indicators used to clinically diagnose perimenopausal syndrome. The peripheral blood FSH in perimenopausal women is approximately 14 times higher than that of normal young women, and the LH levels are enhanced by approximately three-fold [42]. These hematological parameters can be used to predict the occurrence of metabolic syndrome in perimenopausal and postmenopausal women [43]. The LH levels are, to a certain extent, related to the activation of the hypothalamic–pituitary–gonadal axis [44]. Studies have shown that ovarian dysfunction and changes in E_2_, T, FSH, and LH are an expression of kidney deficiency [45]. In the present study, we also found that the serum levels of E_2_, T in the model group were decreased, and the levels of FSH and LH were increased. Icariin administrated for 30 days significantly increased the levels of E_2_ and T, and decreased the levels of FSH and LH, restoring the disordered hormones to normal levels.

IL-2 is an important cytokine in the immune system, and is mainly produced by T cells. E2 can affect the release of immune mediators, including IL-2, via its receptors in immunocompetent cells [46,47]. Numerous studies have shown that during the perimenopausal period, T lymphocyte function declines, and IL-2 levels and activity also decrease. The primary function of IL-2 is to enhance the immune response [48]. A decrease or increase in the quality of immune viscera (visceral indices) represents immunosuppressive or immune-boosting states, respectively [49]. The determined visceral indices of the thymus and spleen indicated their degree of atrophy. Our preliminary study showed that icariin could improve the apparent symptoms of perimenopausal depression, increase the visceral indices of the thymus and spleen, as well as IL-2 levels, and boost immune function in the model rats.

The neurotransmitter theory is a widely accepted view of the pathogenesis of perimenopausal depression. It is believed that the occurrence of perimenopausal depression may be associated with complicated interactions among neurotransmitters. Changes in the levels of brain neurotransmitters such as NE, DA, and 5-HT indicate an imbalance in brain monoamine transmitters, which results in depression. Studies have demonstrated that by adjusting the levels of 5-HT and DA, antidepressant effects can be achieved [50,51,52,53]. Saikosaponin was shown to reduce hippocampal neuroinflammation and restore immune response in a perimenopausal depression model by mediating the neuroendocrine system, resulting in antidepressant effects [54]. Studies have also shown that the significantly reduced E_2_ levels in female patients with menopausal depression and the resulting decreased expression levels of 5-HT, DA, and NE in the brain are the core factors in the pathogenesis of menopausal depression in women [55]. In our study, icariin also increased the levels of the monoamine neurotransmitters 5-HT, DA, and NE in brain homogenates, relieving the depression in model rats.

The PI3K–AKT signaling pathway is intact in oocytes and granulosa cells, which coordinately regulates the growth, development, and maturation of follicles [56]. It has been reported that regulation of the expression of PI3K–AKT signaling molecules in the ovaries improves granulosa cell survival and ovarian follicle development [57]. Acupuncture therapy may upregulate PI3K-AKT signaling pathway-related genes and protein expression; therefore, it could also treat premature ovarian dysfunction with comparable efficacy to estrogen [58].

In addition, the PI3K/AKT signaling pathway is closely associated with ERα activity, with each regulating the other [59,60]. In the absence of estrogen, the binding of ERα to the regulatory subunit p85α of PI3K initiates the PI3K–AKT pathway; p-AKT is thought to phosphorylate ser167 of ERα, suggesting an ERα-PI3K/AKT2-ERα pathway [61]. Santen et al. studied the recurrence of breast cancer after the patients went through oophorectomy [62], and Bratton et al. studied the regulatory effects of ER/PI3K-AKT on the survival of MCF-7 breast cancer cells [63]. It was found that by activating AKT, ERα caused an enhanced transcription of the Bcl-2 gene promoter and eventually led to increased Bcl-2 protein levels, indicating that ER–PI3K–AKT is a complete signaling pathway. This effect occurred in both the absence and presence of estrogen, suggesting that this signaling pathway improved the proliferation and differentiation of ovarian cells, and simultaneously interacted with ERα to improve estrogen levels, which may in turn improve perimenopausal syndrome. The present study found that icariin promoted the expression of ER and ERα in the hypothalamus, increased estrogen levels, encouraged the expression of Bcl-2, AKT, p-AKT, PI3K (110 kDa), and PI3K (85 kDa) in the ovaries, and inhibited the ovarian expression of Bax. Taking the findings of the present study, it is reasonable to suggest that there is a regulatory effect of icariin on PI3K–AKT signaling. We speculated that the antidepressant-like effects of icariin might be a result of the PI3K–AKT signaling activation in perimenopausal depression rats.

## 4. Materials and Methods

### 4.1. Animals

Female SD rats weighing 250–270 g were purchased from the Ji’nan Peng Yue experimental animal breeding Co.Ltd. (Jinan, China). Animals were single-housed under standard laboratory conditions of food and water ad libitum, 22 + 2 °C, a 12-h light:dark cycle (lights on at 08:00) and relative humidity 50–60% unless otherwise specified. All experiments were conducted in accordance with guidelines of Animal Care and Use Committee of Henan University of Chinese Medicine. Every effort was made to minimize the number of animals used and their suffering.

### 4.2. Drugs and Chemicals

Estradiol Valerate Tablets (Prognova) were purchased from DELPHARM LILLE S.A.S (DELPHARM LILLE S.A.S, Lille, France). The icariin was provided by the chemical chamber of Henan University of Traditional Chinese Medicine. High-performance liquid chromatography (HPLC) was used to determine the content of 98% (Figure 9). The anti-β-actin antibody was purchased from Santa Cruz (Santa Cruz, CA, USA). The antibodies for PI3Kp85, PI3Kp110, and p-Akt were purchased from Cell Signaling Technology (Beverly, CA, USA). The Rat E_2_ ELISA Kit, Rat FSH ELISA Kit, Rat LH ELISA Kit, Rat T ELISA Kit, Rat IL-2 ELISA Kit, Rat 5-HT ELISA Kit, Rat DA ELISA Kit, and Rat NE ELISA Kit were purchased from Suzhou Calvin Biotechnology Co. Ltd. (Suzhou Calvin Biotechnology Co.Ltd., Suzhou, China).

### 4.3. Experimental Design

As depicted in Figure 10, the model preparation method was based on our previous study [10]. Eighty female SD rats with an individual weight of 250–270 g were selected. Among these, 10 rats were randomly assigned to the SG, and the remaining rats were used for model construction by surgery. Following a recovery period of seven days after surgery, to monitor the perimenopausal state of the rats, we observed their estrous cycle through the microscopic examination of vaginal smears.

A total of 50 rats that were successfully perimenopausal modeled and in good condition were selected. The rats were randomly divided into five groups (n = 10). They were the MG, EVG, IG-HD, IG-MD, and IG-LD. The dose of EVG was 0.167 mg/kg. The doses of IG-HD, IG-MD, and IG-LD were 50 mg/kg, 25 mg/kg, and 12.5 mg/kg. All drugs were dissolved in 0.5% carboxymethyl cellulose (CMC). The SG and MG were administered the same volume of 0.5% CMC via intragastric injection. All groups were intragastrically administered with corresponding drugs once daily for 30 consecutive days (Figure 10B). The administration volume was 1 mL/100 g. The rats were weighed every seven days, and the dosage was weight-adjusted. Five days after drug administration, the rats were single caged and administered one CUMS per day for 18 consecutive days.

All animals were subjected to behavioral tests: the open field test followed by the forced swimming test. To further exploit their behavior and avoid multiple behavioral testing on one day, open field test and forced swimming test alternation was assessed one and two days after the last CUMS, respectively.

At the end of the treatment, the animals were anesthetized with 10% chloral hydrate (0.3 mL/100 g, i.p.) [64] and blood samples were collected from the abdominal aorta in nonheparinized tubes for serum separation to estimate E_2_, LH, FSH, T, and IL-2 levels. Subsequently, brains were rapidly dissected and washed with ice-cold saline. Then, each brain was divided into two parts from the sagittal plane. One part of the brains was fixed with 10% (*v*/*v*) formalin for 24 h to perform histopathological staining with immunohistochemistry for the expression of ER and ERα. The other part of the brain was homogenized in ice-cold physiological saline to prepare a 10% homogenate for the assessment of 5-HT, DA, and NE levels. Then, the ovaries were rapidly dissected in the same way as the brain: one part of the ovaries was fixed with 10% (*v*/*v*) formalin for 24 h to perform histopathological staining with hematoxylin and eosin (HE) or immunohistochemistry for the assessment of Bax and Bcl-2 levels; the other part was analyzed for the expression of PI3K, AKT, and P-AKT proteins. The uterus, spleen, and thymus were rapidly extracted for the determination of visceral indices.

Determination methods: ELISA was used to determine the levels of E_2_, LH, FSH, T, IL-2, 5-HT, DA, and NE. Immunohistochemistry was performed to determine the expression of ER, ERα, Bax, and Bcl-2 in the ovaries using a light microscope to observe the average optical. Western blotting was carried out to determine the expression levels of PI3K (110 kDa), PI3K (85 kDa), AKT, and p-AKT proteins.

### 4.4. Surgery 

After weighing, 10% chloral hydrate was intraperitoneally injected at a volume of 0.3 mL/100 g in each rat. The rats were anesthetized and subsequently fixed in the prone position. Under the bottom rib on the rat’s back, approximately 1 cm from the midaxillary line and the outside of the spine, the fur was removed, and the skin was wiped with iodine, followed by ethanol. In the lower third of the trunk, 2 cm from the spine, a longitudinal section was carefully performed with a length of 0.5–1 cm. Ligation of the lower part of both ovarian tubules (including partial fat) was performed. Further, the right ovary and 80% of the left ovary were removed. The tissues were gently pushed back into the abdominal cavity, and the muscle and skin layers were closed separately [65]. In the SG rats, only the incision (sham surgery) was executed, and both ovaries were retained. The postoperative rats were placed in a warm place, carefully fed, and intramuscularly injected with penicillin (200 KU/kg, 0.1 mL/d/rat) for three consecutive days to prevent infection.

### 4.5. CUMS

The CUMS procedure was obtained and followed from previous literature (Table 1) [66]. To prevent habituation and to ensure the unpredictability of the stressors, all stressors were randomly scheduled over a six-day period, and were repeated throughout 18 days. The same stressor was not used for two days, after which animals were returned to their cages. The SG rats were housed in a separate room and had no contact with the stressed animals. Behavioral tests were performed after drug administration, after which animals were sacrificed for further analysis.

### 4.6. Behavioral Assessments

Open field test: To assess locomotor activity, the rats were subjected to a 5-min open field test immediately. Each animal was placed in a black 625 mm (L) × 740 mm (W) × 510 mm (H) video-recorded square box. The videos were analyzed by the opening activity experimental system of OFT-100 software (Techman, Chengdu, China), and the stand-up time and distance moved (cm) by each rat was retrieved. The black box was cleaned after each rat was tested to eliminate possible bias due to odors left by previous rats.

Forced swimming test: Swimming sessions were conducted by placing individual rats in a cylindrical container (50 cm (H) × 21 cm (D) filled with water about 25 °C to a level of 40 cm. On the first session (day 31), rats were allowed to swim for 15 min. 24 h later (day 32), rats were again placed in the cylinder for 5 min. Time spent on immobility (when no additional movements other than those necessary to keep the rat’s head out of the water) was recorded. Water was replaced between every trial. At the end of the experiment, all rats were allowed to dry in a heated enclosure and placed back in their home cages.

### 4.7. Western Blot Analysis 

Ovaries levels of the PI3K, AKT, and p-AKT proteins were analyzed using the Western blot methodology [67]. Afterwards, proteins were extracted from ovaries tissues by a protein extraction kit. The equal amounts of proteins were separated by SDS-polyacrylamide gel electrophoresis and transferred to polyvinylidene difluoride membranes (PVDF). Then, the membranes were blocked with 5% (w/v) skim milk in Tris-buffered saline containing 0.05% Tween 20 (TBST) at room temperature with constant shaking. Two hours later, the membranes were incubated with a 1:1000 dilution of antibodies against rat p-AKT, PI3K, and β-actin for 12 h at 4 °C. Next, the membranes were washed three times for 10 min, each time with TBST, and were then incubated with a horseradish peroxidase-conjugated anti-rabbit secondary antibody for 1 h at room temperature. After washing, the membrane was immersed in luminescent liquid (prepared before use), incubated at room temperature for 3 min, and photographed after film exposure. The images were subsequently subjected to densitometric analysis.

### 4.8. Histopathological Examination and Immunofluorescence

The paraffin section procedure: After ovaries and brains were fixed with 10% formalin for 24 h, the samples were dehydrated by incubations in different concentration of alcohol. Then, they were cleared with xylene. Afterwards, they were embedded in paraffin at 56 °C in a hot air oven for 24 h. Then, ovaries and coronal brain sections were processed for paraffin embedding, and 4-μm sections were prepared.

Histopathological Examination: Ovary sections were then stained with HE and examined under a light microscope (Olympus BX61, Tokyo, Japan). A scoring system ranging from 0 to 3 points was used to evaluate the degree of severity of the observed histopathological changes, where 3 indicates that there are no folliculi in ovary, a lot of corpus luteum, ovarian atrophy, and no blood vessels. A score of 2 indicates that mature follicles and the corpus luteum can be seen in ovarian tissue, a lot of corpus luteum, the number of layers of granulosa cells is less, and there are a few blood vessels. A score of 1 indicates that the growing follicle, mature follicle, and corpus luteum are visible in the ovarian tissue, while the mature follicle, corpus luteum, and the number of layers of granulosa cells is less. Lastly, a score of 0 indicates that ovarian follicles, mature follicles, and corpus luteum are seen in the ovarian tissue. The corpus luteum is well developed, granular cells have many layers, and the follicular fluid and blood vessels are rich [65].

Immunofluorescence: Every fourth ovary and brain section was incubated with a blocking buffer for 1 h. Then, the sections were incubated in 4% PFA for 48 h at 4 °C. After washing with PBS, the sections were immersed in 0.3% PBST for 10 min. The sections were blocked and incubated with anti-ER or anti-ER α (Abcam, MA, USA) overnight. Afterwards, the sections were washed with PBS and then incubated with a secondary antibody for 1 h at room temperature. After washing, sections were then stained with diaminobenzidine (DBA) (Thermofisher, MA, USA) and examined under a light microscope (Olympus BX61, Tokyo, Japan) performed to determine the expression of ER, ERα, Bax, and Bcl-2. Positive staining shows different degrees of yellow or pale brown [68].

### 4.9. Enzyme-Linked Immunosorbent Assay

The levels of E_2_, LH, FSH, T, IL-2, 5-HT, DA, and NE were estimated using rat ELISA kits purchased from Suzhou Calvin Biotechnology Co. Ltd. (Suzhou Calvin Biotechnology Co. Ltd., China). The procedures were performed according to the manufacturer’s instructions. Coronal brain sections were processed for paraffin embedding, and 4-μm sections were prepared.

### 4.10. Data Analysis

SPSS 21.0 software was used for the statistical analysis of the results. The data are presented as means ± S.D. One-way ANOVA was used for comparison among groups. The Games–Howell test was used for those with uneven variance, the LSD test was used for those with even variance, and the Ridit test was used for ranked data. GraphPad Prism software (version 6, GraphPad Software, Inc., San Diego, CA, USA) was used to create the graphs.

## 5. Conclusions

In conclusion, this study showed that icariin (50 mg/kg, 25 mg/kg) improved depression symptoms in the model rats via regulation of the PI3K-AKT pathway, boosting the expression of the regulatory methylene p85 and catalytic methylene p110 of PI3K, increasing the relative expression of p-AKT, and encouraging the expression of the anti-apoptosis factor Bcl-2, thus elevating the ratio of Bcl-2/Bax. In addition, p-AKT promoted the expression of ERα, and the two interacted with each other to improve ovarian function, thereby regulating hormone levels and immune function in the model rats. The mechanistic details need to be further investigated.

## Figures and Tables

**Figure 1 molecules-24-03700-f001:**
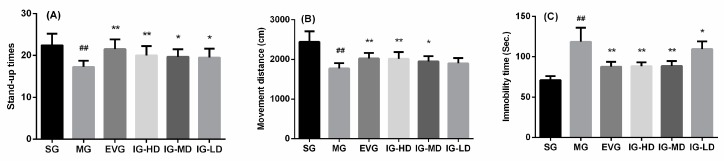
Icariin administration for 30 days increased the stand-up times (**A**) and movement distance (**B**) in the open field test and decreased immobility time (**C**) in the forced swimming test in perimenopausal depression rats. Sham group (SG), model group (MG), estradiol valerate group (EVG, 0.167 mg/kg), high-dose icariin group (IG-HD, 50 mg/kg), medium-dose icariin group (IG-MD, 25 mg/kg), low-dose icariin group (IG-LD, 12.5 mg/kg). Values are expressed as mean ± S.D, *n* = 10. ## *p* < 0.01 versus the SG. ** *p* < 0.01 versus the MG. * *p* < 0.05 versus the MG. (Statistical analyses were performed using ANOVA, and the criterion for statistical significance was set to *p* < 0.05).

**Figure 2 molecules-24-03700-f002:**
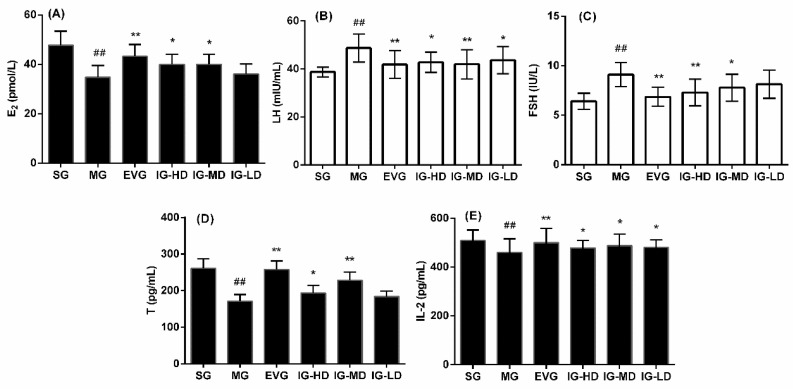
Icariin administration for 30 days increased the levels of estradiol (E_2_) (**A**), follicle stimulating hormone (FSH) (**C**) and interleukin-2 (IL-2) (**E**). However, icariin could reduce the levels of LH(**B**) and T(**D**) in perimenopausal depression rats. The sham group (SG), model group (MG), estradiol valerate group (EVG, 0.167 mg/kg), high-dose icariin group (IG-HD, 50 mg/kg), medium-dose icariin group (IG-MD, 25 mg/kg), and low-dose icariin group (IG-LD, 12.5 mg/kg). Values are expressed as mean ± S.D, *n* = 10. ## *p* < 0.01 versus the SG. ** *p* < 0.01 versus the MG. * *p* < 0.05 versus the MG. (Statistical analyses were performed using ANOVA, and the criterion for statistical significance was set to *p* < 0.05).

**Figure 3 molecules-24-03700-f003:**
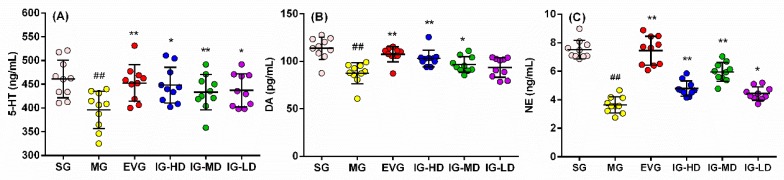
Icariin administration for 30 days improve the levels of serotonin (5-HT) (**A**), dopamine (DA) (**B**) and noradrenaline (NE) (**C**) in perimenopausal depression rats. Sham group (SG), model group (MG), estradiol valerate group (EVG, 0.167 mg/kg), high-dose icariin group (IG-HD, 50 mg/kg), medium-dose icariin group (IG-MD, 25 mg/kg), and low-dose icariin group (IG-LD, 12.5 mg/kg). Values are expressed as mean ± S.D, *n* = 10. ## *p* < 0.01 versus the SG. ** *p* < 0.01 versus the MG. * *p* < 0.05 versus the MG. (Statistical analyses were performed using ANOVA, and the criterion for statistical significance was set to *p* < 0.05).

**Figure 4 molecules-24-03700-f004:**
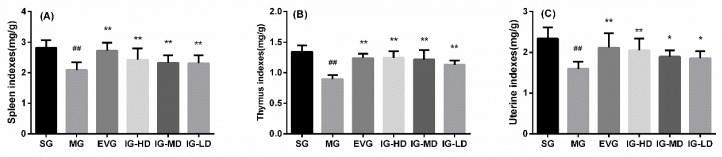
Icariin administration for 30 days ameliorated the indexes of spleen (**A**), thymus (**B**), and uterine (**C**) in perimenopausal depression rats. Values are expressed as mean ± S.D, n = 10. ## *p* < 0.01 versus the SG. ** *p* < 0.01 versus the MG. * *p* < 0.05 versus the MG. Sham group (SG), model group (MG), estradiol valerate group (EVG, 0.167 mg/kg), high-dose icariin group (IG-HD, 50 mg/kg), medium-dose icariin group (IG-MD, 25 mg/kg), low-dose icariin group (IG-LD, 12.5 mg/kg). Values are expressed as mean ± S.D, *n* = 10. ## *p* < 0.01 versus the SG. ** *p* < 0.01 versus the MG. * *p* < 0.05 versus the MG. (Statistical analyses were performed using ANOVA, and the criterion for statistical significance was set to *p* < 0.05).

**Figure 5 molecules-24-03700-f005:**
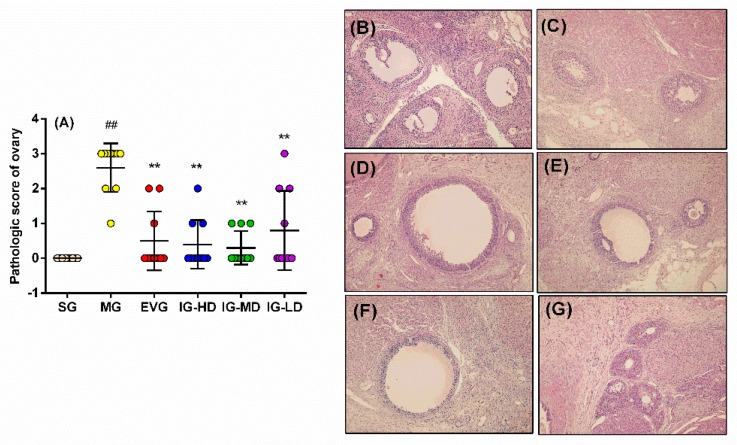
Icariin administration for 30 days improves the ovary pathological changes (**A**) in perimenopausal depression rats. Sham group (SG), model group (MG), estradiol valerate group (EVG, 0.167 mg/kg), High-dose icariin group (IG-HD, 50 mg/kg), medium-dose icariin group (IG-MD, 25 mg/kg), low-dose icariin group (IG-LD, 12.5 mg/kg). Values are expressed as mean ± S.D, n = 10. ## *p* < 0.01 versus the SG. ** *p* < 0.01 versus the MG. * *p* < 0.05 versus the MG. Representative photomicrographs of SG (**B**), MG (**C**), EVG (**D**), IG-HD (**E**), IG-MD (**F**), IG-LD and (**G**) (HE × 200). (Statistical analyses were performed using the Ridit test, and the criterion for statistical significance was set to *p* < 0.05).

**Figure 6 molecules-24-03700-f006:**
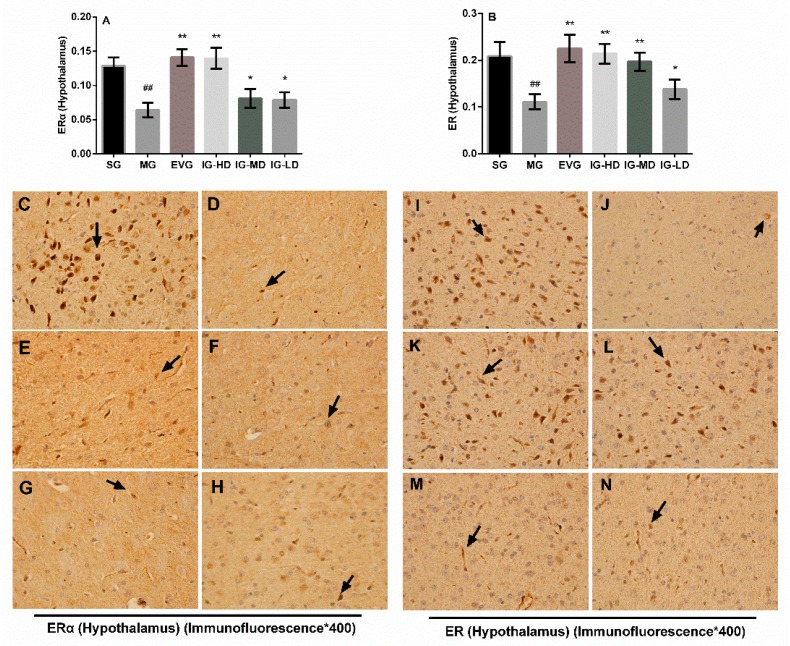
Icariin administration for 30 days increased the expression of ER (**A**) and ERα (**B**) in the hypothalamus of perimenopausal depression rats. Sham group (SG), model group (MG), estradiol valerate group (EVG, 0.167 mg/kg), high-dose icariin group (IG-HD, 50 mg/kg), medium-dose icariin group (IG-MD, 25 mg/kg), low-dose icariin group (IG-LD, 12.5 mg/kg). Values are expressed as mean ± S.D, *n* = 10. ## *p* < 0.01 versus the SG. ** *p* < 0.01 versus the MG. * *p* < 0.05 versus the MG. Representative photomicrographs of SG (C/I), MG (D/J), EVG (E/K), IG-HD (F/L), IG-MD (G/M), IG-LD (H/N) (immunofluorescence ×400). (Statistical analyses were performed using ANOVA, and the criterion for statistical significance was set to *p* < 0.05).

**Figure 7 molecules-24-03700-f007:**
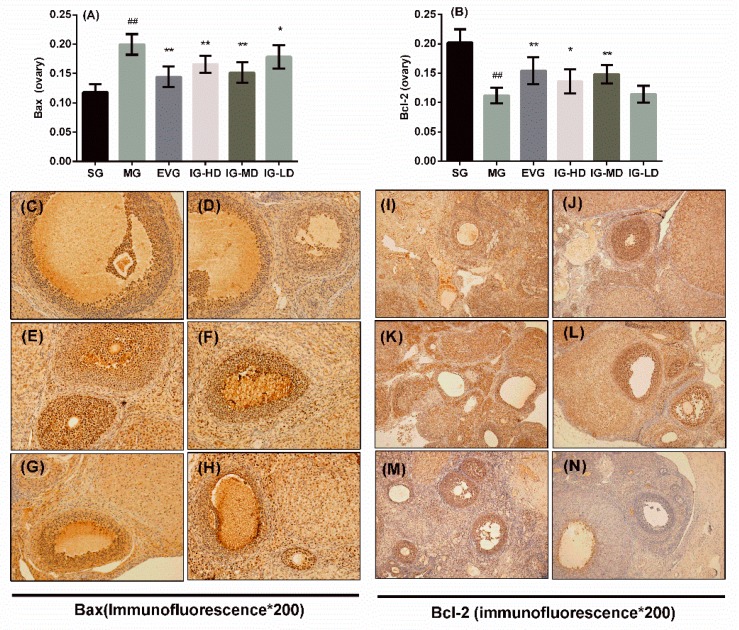
Icariin administration for 30 days increased the expression of bax (Figure 7**A**) and B-cell lymphoma 2 (Bcl-2) (Figure 7**B**) in perimenopausal depression rats. Sham group (SG), model group (MG), estradiol valerate group (EVG, 0.167 mg/kg), high-dose icariin group (IG-HD, 50 mg/kg), medium-dose icariin group (IG-MD, 25 mg/kg), and low-dose icariin group (IG-LD, 12.5 mg/kg). Values are expressed as mean ± S.D, *n* = 10. ## *p* < 0.01 versus the SG. ** *p* < 0.01 versus the MG. * *p* < 0.05 versus the MG. Representative photomicrographs of SG (**C** and **I**), MG (**D** and **J**), EVG (**E** and **K**), IG-HD (**F** and **L**), IG-MD (**G** and **M**), and IG-LD (**H** and **N**) (Immunofluorescence ×400).(Statistical analyses were performed using ANOVA, and the criterion for statistical significance was set to *p* < 0.05).

**Figure 8 molecules-24-03700-f008:**
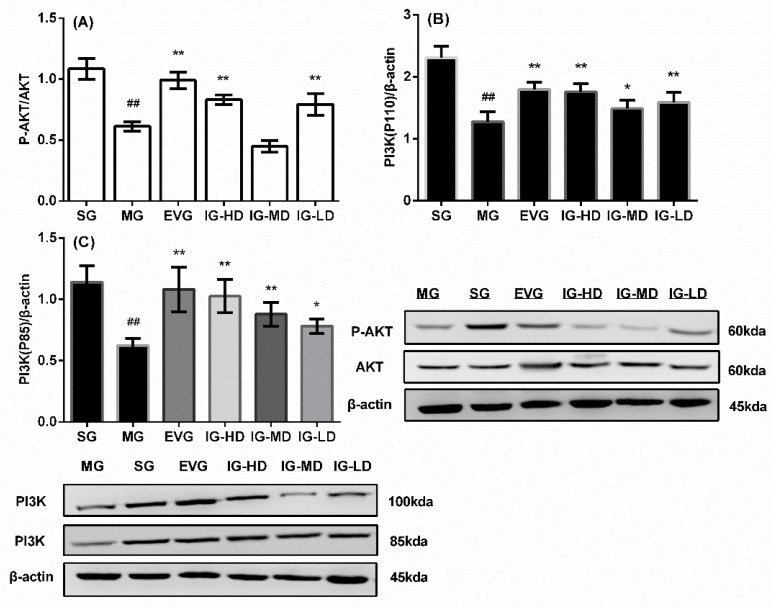
Icariin administration for 30 days increased phosphorylation-serine-threonine protein kinase (P-AKT)/AKT (**A**) and the expression of phosphatidylinositol 3-kinase (PI3K) (110 kd) (**B**) and PI3K (85 kd) (**C**) in perimenopausal depression rats. Sham group (SG), model group (MG), estradiol valerate group (EVG, 0.167 mg/kg), high-dose icariin group (IG-HD, 50 mg/kg), medium-dose icariin group (IG-MD, 25 mg/kg), and low-dose icariin group (IG-LD, 12.5 mg/kg). Values are expressed as mean ± S.D, *n* = 10. ## *p* < 0.01 versus the SG. ** *p* < 0.01 versus the MG. * *p* < 0.05 versus the MG. (Statistical analyses were performed using ANOVA, and the criterion for statistical significance was set to *p* < 0.05).

**Figure 9 molecules-24-03700-f009:**
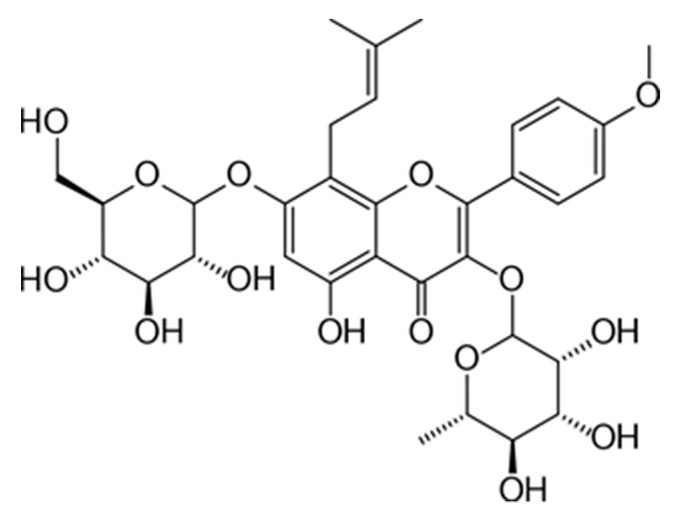
Chemical structure of icariin.

**Figure 10 molecules-24-03700-f010:**
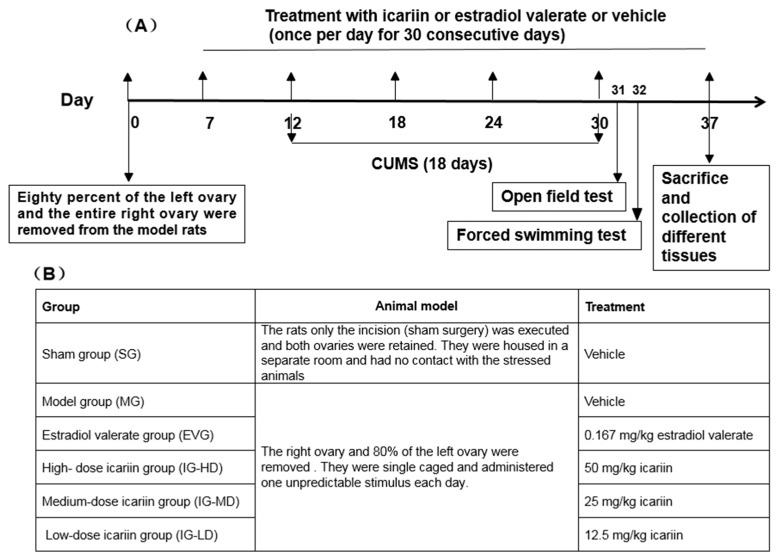
Schematic representation of experimental design (**A**) and overview of treatment groups (**B**).

**Table 1 molecules-24-03700-t001:** Chronic unpredictable mild stress (CUMS) procedure.

Stressor	Details
Food deprivation	Rats were subjected to 24 h of food deprivation. Food was provided immediately after the end of the fasting period.
Water deprivation	Rats were subjected to 24 h of water deprivation. Water was provided immediately after the end of the fasting period.
Overnight illumination	Lights on for 24 h starting 20:30
Wet sawdust bedding	1 mL of water per 1 g of sawdust bedding for 20 h. Immediately after the stress, rats were removed from the cage and towel dried before being placed back in their home cage.
Heat environment	45 °C for 5 min
Cold water swimming	Rats were placed for 5 min in a cylindrical clear plastic tank (46 cm high (H) × 20 cm diameter (D) filled with water (4 ± 1 °C) to a depth of 30 cm. Immediately after the swim, rats were removed from the tank and towel dried before being placed back in their home cage.

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
