# Peer review of "PI3K–AKT Signaling Activation and Icariin: The Potential Effects on the Perimenopausal Depression-Like Rat Model"

_molecules, 2019, doi:10.3390/molecules24203700_

Round 1

Reviewer 1 Report

Having read the manuscript entitled “The antidepressant-like effects of Icariin: Dependent on the PI3K-AKT signaling activation in Perimenopausal depression-like rats” I have to admit that the topic is quite interesting. However, I have several comments:

There is a mistake in the title of the manuscript.

Abstract section: All abbreviations should be explained when they appear for the first time.

Keywords: It should be “perimenopausal depression” instead of “perimenopausal depression-like”. I would also add “rats”, “forced swim test”, “open field test”

The body of the manuscript: All abbreviations should be explained when they appear for the first time in the text.

Page 2, line 45: Estrogen replacement therapy cannot be referred to as “a supplementation”. Such a term is definitely a too far-reaching simplification.

Page 2, line 55: A reference should be added.

Page 2, The Introduction section lacks a clearly defined aim of the study.

Page 2, line 78: It should be “IG-MD” instead of “IG-HD”.

Page 5, line 156: It should be “increased” instead of “decreased”.

Page 8, line 213, page 9, line 235: It should be “30 days” instead of “4 weeks”.

Result section: According to point 2.1. and Fig.1, EVG and icariin elevated locomotor activity of animals (observed as an increased distance travelled by a given rat), which can confound the results obtained in the forced swim test. How the Authors are certain that the reduced immobility time in the FST were not a result of increased locomotion of animals?

In the Discussion section the Authors should shortly explained why Estradiol Valerate Tablets were used. Since not all icariin doses prevented the noxious effects of surgery and CUMS, it is not enough to state that for example “icariin increased the levels of the monoamine neurotransmitters” – the active dose/doses should be specified.

Page 10, line 289-290: Authors should ensure that their research complies with the commonly-accepted “3R” rule and that it was conducted in accordance with relevant national legislation on the use of animals for research. University guidelines are not enough.

Page 11, line 320: It should be “SG” instead of “NG”

Page 11, line 331: an appropriate reference for chloral hydrate anesthesia should be given, which confirms that administration of this agent does not influence the performed tests.

Page 12, point 4.4 lacks a reference.

Page 13, lines 369-381 should be removed, since this information is repeated in lines 383-395.

Page 13, point 4.8 lacks a reference.

Page 14, the immunofluorescence method lacks a reference.

Figure captions: All abbreviations that appear in figures should be explained in figure captions. Route of administration, administered doses, and a name of the applied statistical test should be given in figure captions.

The manuscript needs additional English revision and careful correction of typos.

Reviewer 2 Report

First, this paper has reported a novel effect for icariin for the postmenopausal depression-like disease in a rat model. However, there are several errors in spelling and grammar, for examples:

In the Abstract:

Recently, the antidepressant-like mechanism of Icariin has been increasingly evaluated and de-nstrated (demonstrated?).

Perimenopausal depression-like (disease? disorder? effects?) is a chronic recurrent disease, which leads to increased risk of suicide, and poses a significant risk to public health.

Actually, these errors spread through the manuscript. An over-all English editing is needed.  

I found that the main theme of this article is the potential of a TCM herb-derived chemical in the treatment of postmenopausal depression-like disease, evaluated by a rodent model. Therefore, I suggest change the Title as:

“PI3K-AKT signaling activation and Icariin: the potential effects on the Perimenopausal depression-like rat model”

You mentioned in the Introduction as “Perimenopausal depression is a common emotional disorder caused by an ovarian dysfunction-induced decrease in estrogen and psychosocial factors, and the incidence rate is rising.” Why don’t you mention this issue further as estrogen drop during the perimenopausal phase? Basically, the authors have established a new model of the postmenopausal depression-like disease and then evaluated the effects of icariin in this model. However, there is lack of a statement or detailed description about the validity of this model for postmenopausal depression. That is, authors are advised to compare this new model and previous rat models for depression and postmenopausal dep

Round 2

Reviewer 1 Report

All my suggestions were taken into consideration.

Reviewer 2 Report

The authors have addressed all the points that I have mentioned in the previous review, I suggest accept the article.